# Bridging cell-scale simulations and radiologic images to explain short-time intratumoral oxygen fluctuations

**Jessica L. Kingsley**[1,2], **James R. Costello**[3], **Natarajan Raghunand**[4,5], **Katarzyna A. Rejniak**[1,5]*

**1** Department of Integrated Mathematical Oncology, H. Lee Moffitt Cancer Center & Research Institute, Tampa, Florida, United States of America, **2** University of South Florida, Tampa, Florida, United States of America, **3** Department of Radiology, H. Lee Moffitt Cancer Center & Research Institute, Tampa, Florida, United States of America, **4** Department of Cancer Physiology, H. Lee Moffitt Cancer Center & Research Institute, Tampa, Florida, United States of America, **5** Department of Oncologic Sciences, Morsani College of Medicine, University of South Florida, Tampa, Florida, United States of America

\* Kasia.Rejniak@moffitt.org

**Data Availability Statement:** The computational code is deposited at GitHub: https://github.com/rejniaklab/MultiCell-O2-fluctuations.

**Funding:** This work was supported in part by the U01-CA202229 Physical Sciences Oncology

## Abstract

Radiologic images provide a way to monitor tumor development and its response to therapies in a longitudinal and minimally invasive fashion. However, they operate on a macroscopic scale (average value per voxel) and are not able to capture microscopic scale (cell-level) phenomena. Nevertheless, to examine the causes of frequent fast fluctuations in tissue oxygenation, models simulating individual cells' behavior are needed. Here, we provide a link between the average data values recorded for radiologic images and the cellular and vascular architecture of the corresponding tissues. Using hybrid agent-based modeling, we generate a set of tissue morphologies capable of reproducing oxygenation levels observed in radiologic images. We then use these in silico tissues to investigate whether oxygen fluctuations can be explained by changes in vascular oxygen supply or by modulations in cellular oxygen absorption. Our studies show that intravascular changes in oxygen supply reproduce the observed fluctuations in tissue oxygenation in all considered regions of interest. However, larger-magnitude fluctuations cannot be recreated by modifications in cellular absorption of oxygen in a biologically feasible manner. Additionally, we develop a procedure to identify plausible tissue morphologies for a given temporal series of average data from radiology images. In future applications, this approach can be used to generate a set of tissues comparable with radiology images and to simulate tumor responses to various anticancer treatments at the tissue-scale level.

## Author summary

Low levels of oxygen, called hypoxia, are observable in many solid tumors. They are associated with more aggressive malignant cells that are resistant to chemo-, radio-, and immunotherapies. Recently developed imaging techniques provide a way to measure the magnitude of frequent short-term oxygen fluctuations, but they operate on a macro-scale

Project (PS-OP) grant from the US National Institutes of Health, National Cancer Institute (to KR), the Moffitt Radiology Pilot Project grant (to JC), and Shared Resources at the H. Lee Moffitt Cancer Center & Research Institute an NCI designated Comprehensive Cancer Center under the grant P30-CA076292 from the National Institutes of Health (Moffitt IRAT Core, to NR). The funders had no role in study design, data collection and analysis, decision to publish, or preparation of the manuscript.

**Competing interests:** The authors have declared that no competing interests exist.

voxel level. To examine the possible causes of rapid oxygen fluctuations at the cell level, we developed a hybrid agent-based mathematical model. We tested two different mechanisms that may be responsible for these cyclic effects on tissue oxygenation: temporal variations in vascular influx of oxygen and modulations in cellular oxygen absorption. Additionally, we developed a procedure to identify plausible tissue morphologies from data collected from radiological images. This can provide a bridge between the micro-scale simulations with individual cells and the longitudinal medical images containing average values. In future applications, this approach can be used to generate a set of tissues compatible with radiology images and to simulate tumor responses to various anticancer treatments at the cell-scale level.

## Introduction

Tumor tissues harbor regions of different levels of oxygen, including the well-oxygenated areas (normoxia) and zones with reduced oxygen availability (hypoxia). The hypoxic regions can arise as a result of rapid proliferation of tumor cells and tortuous tumor vasculature, which together lead to an increased distance between some tumor cells and the nearest blood vessel. This, in turn, results in the emergence of oxygen gradients and a diffusion-limited hypoxia, usually at the distances of 120–180 μm from vasculature[1]. Such chronic hypoxia, with oxygen partial pressure ($pO_2$) below 10 mmHg, may last for a prolonged periods of time, often for more than 24 hours[2,3]. However, hypoxic regions can also be created due to irregular blood flow in the aberrant tumor vasculature or shutdown of small vessels. These phenomena lead to a perfusion-limited hypoxia that is observable for shorter times, often minutes to hours, and can be reversed when the blood flow is restored[2,3]. Several studies have demonstrated the existence of 20–30 minute-long cycles in red blood cell flux that can vary by 2–5 fold and lead to periodic changes in $pO_2$ within the tumor tissue[3–5]. However, it has also been observed in murine experiments that tumors experience very fast and sometimes quite large fluctuations (more than 5-fold) in oxygen levels within the tissue. In particular, electron paramagnetic resonance (EPR) imaging has shown that intratumor fluctuations in $pO_2$ can reach a magnitude as high as 30 mmHg over a period as short as 3 minutes[6,7].

EPR imaging is a spectroscopic technique that can detect molecules presenting unpaired electrons. However, viable tissues contain insufficient amounts of radical species, so EPR requires the injection of a paramagnetic probe to enable visualization of $pO_2$[8]. One such a probe, the triaryl-methyl (TAM) radicals, are injected intravascularly to serve as a tracer for mapping and quantifying tissue $pO_2$ in live animals[6,7]. As a result of collisions between TAM and $O_2$ molecules, the TAM spectral line width broadens in proportional to $pO_2$ and can be detected by the EPR scanner. The subsequent oxygen image reconstruction provides quantitative maps of $pO_2$ distribution within the tissue[9]. The typical EPR voxel has a resolution of $1mm^3$, and average $pO_2$ values from each volumetric voxel are collected in the form of 2D oxygenation maps (compare Fig 2 in [6]). The EPR oxygenation maps show spatial variations in the xy-plane, but are not able to capture variations in the third dimension.

This imaging technique was used to record intratumoral $pO_2$ fluctuations in the squamous cell carcinoma VII (SCCVII) of size 1,200 $mm^3$ shown in Fig 2 of [6]. In this experiment, four regions of interest (ROIs) were selected based on an anatomical image of the tumor from the T2-weighted magnetic resonance imaging (MRI). EPR imaging was used to record $pO_2$ maps that were co-registered with the MRI image and average values of $pO_2$ in each ROI were recorded every 3 minutes for 24 minutes. The $pO_2$ maps correspond to the last data set (at 28 minutes). The four chosen ROIs are characterized by different initial levels of $pO_2$, from a very

well oxygenated Region #1 to a severely hypoxic Region #4. Regions #1 and #2 showed significant changes in $pO_2$ during the time of experiment (more than 10-fold), while Regions #3 and #4 displayed more uniform levels of $pO_2$ during the whole experiment. Since the EPR imaging operates on the resolution of millimeters (macroscale), it is impossible to determine which biological mechanisms are responsible for such fast and relatively large (more than 5-fold) oxygen fluctuations. To address this issue, the modeling on a cellular level (microscale) is needed.

Motivated by these experimental data, we use the hybrid agent-based *MultiCell-LF* (multi-cell lattice-free[10,11]) model to test mechanisms that could be responsible for cyclic effects in tissue oxygenation. In general, the distribution of blood-borne compounds (oxygen, nutrients, drugs, etc.) within the tissue depends on the localization and amount of compound entering the tissue (vascular supply), the amount and localization of compound leaving the tissue (cellular uptake), and compound interstitial transport. We consider here the first two mechanisms only. Since the experiments were performed over a very short period of time (30 minutes), no significant modifications in oxygen transport were expected. Such modifications can arise only due to changes in the number or localization of cells or vessels, or if the extracellular matrix composition changes. In a half hour period and with no therapeutic interventions (such as radiotherapy or surgery), no changes in the number of cells (no proliferation, no death), in the vasculature (no angiogenesis, no vascular collapse due to tumor growth), nor in the ECM structure (no ECM production by stromal cells) are anticipated. Additionally, in the experiments described above, the levels of an EPR-specific tracer (TAM) were recorded, but no detectable changes in tracer levels or distribution were observed within each ROI (compare Fig 2 from[6]). This also suggests that the observed oxygen fluctuations are not related to changes in the interstitial transport. Therefore, we tested whether the modifications in vascular oxygen supply or in cellular oxygen absorption can contribute to the observed variations in tissue oxygenation.

The rest of the paper is organized as follows. The mathematical model is described in **section 2** and used to design a collection of tumor tissues (**section 3.1**) with a numerically stable oxygen distribution (**section 3.2**). Next, for four tissues that match the average $pO_2$ value in each experimental ROI (**section 3.3**), we determine optimal rates of oxygen vascular influx or oxygen cellular uptake that fit the experimentally observable fluctuations in $pO_2$ (**section 3.4**). These optimal schedules are then applied to a larger sample of *in silico* tissues with initial $pO_2$ values close to the experimental data to assess schedules' robustness and reproducibility (**section 3.5**). Finally, we discuss implications of our findings for tumor development and future applications (**section 4**).

## Methods–mathematical model

For this study, we consider a two-dimensional tissue patch with an area of 1 mm$^2$ that corresponds to a cross section of a typical EPR imaging voxel and a single element in the EPR oxygen map[12,13]. Tissue morphology and metabolism is modeled using the *MultiCell-LF* model, which combines the off-lattice individual vessels and cells (tumor and stromal) with a continuous description of oxygen kinetics. A typical example from our simulations is shown in **Fig 1**. The specific oxygen distribution within the tissue depends on three factors: the amount of oxygen supplied from individual vessels, the amount of oxygen absorbed by both tumor and stromal cells, and the spatial localization of all cells and vessels. As a result of this influx-outflux balance, an irregular oxygen distribution pattern can emerge.

### Tissue design

Initially, the locations of tumor cells, stromal cells and vessels were chosen randomly within the tissue domain. To ensure that the cells did not overlap with one another and with the vessels, repulsive forces were applied to all cells. Let $X_i$ and $X_j$ represent the coordinates of two

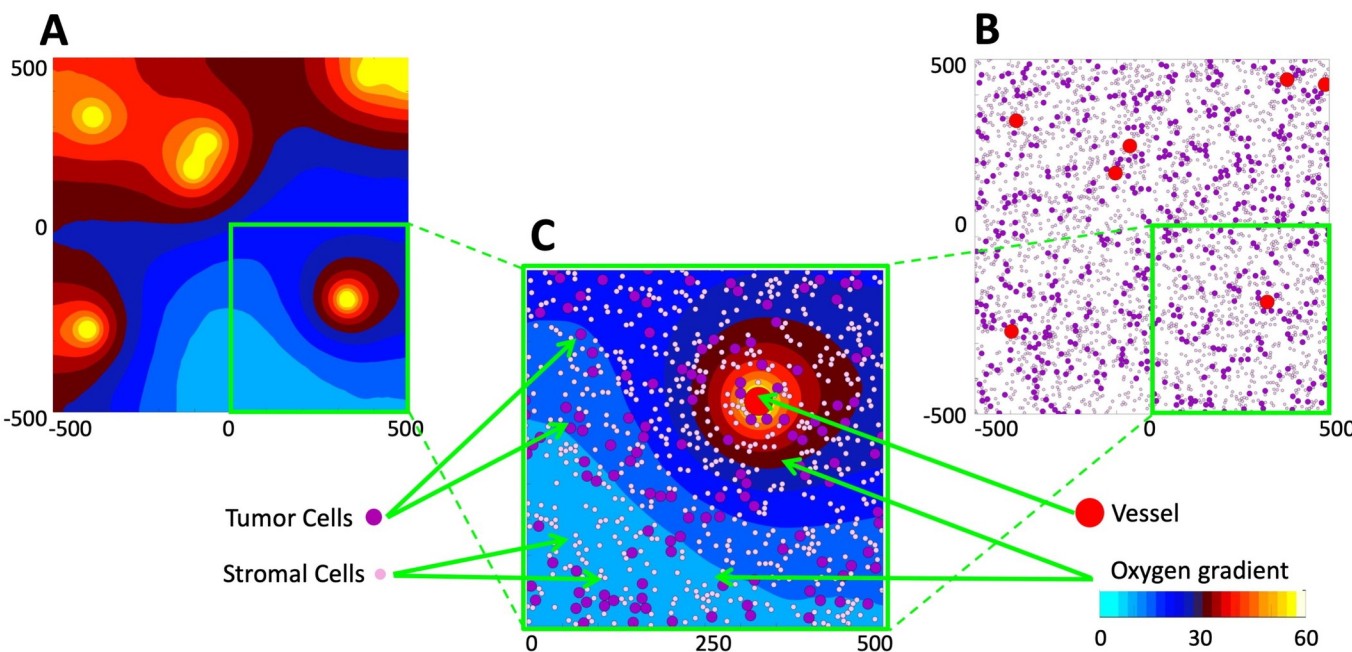

**Fig 1. Mathematical model of the tumor tissue microenvironment. A.** A contour map of the simulated oxygen distribution. The color scheme corresponds to that used in EPR imaging for the partial pressure of oxygen (cyan: low pO$_2$; white: high pO$_2$). **B.** Locations of tumor vasculature (red circles), tumor cells (purple circles), and stromal cells (pink circles) within the same computational domain; this is used to define tissue cellularity and vascularity. The color scheme corresponds to typical colors in histology images. **C.** Magnification of a quarter of the computational domain showing all model components together: the vessels, tumor and stromal cells, and oxygen distribution. All variables are dimensional (length is in *μm*, oxygen partial pressure in *mmHg*, as listed in Table 1).

discrete elements (either tumor cells, stromal cells or vessels) of radii $R_i$ and $R_j$, respectively. The repulsive Hookean force $f_{X_i,X_j}$ of stiffness $\mathcal{F}$ acting on element $X_i$ is given by:

$$f_{X_i,X_j} = \begin{cases} \mathcal{F}\Big((R_i + R_j) - \|X_i - X_j\|\Big)\dfrac{X_i - X_j}{\|X_i - X_j\|} & \text{if } \|X_i - X_j\| < R_i + R_j \\ 0 & \text{otherwise.} \end{cases}$$

For the tissue that contains $N_V$ vessels of coordinates $V_i = (V_i^x, V_i^y)$, $N_T$ tumor cells of coordinates $T_j = (T_j^x, T_j^y)$, and $N_S$ stromal cells of coordinates $S_k = (S_k^x, S_k^y)$, the repulsive forces $F_j^T$ acting on tumor cells and $F_k^S$ acting on stromal cells combine contributions from all nearby tumor cells, stromal cells and vessels, and are given by the following equations:

$$F_j^T = \sum_{l \neq j}^{N_T} f_{T_j,T_l} + \sum_{k=1}^{N_S} f_{T_j,S_k} + \sum_{i=1}^{N_V} f_{T_j,V_i} \quad for\ j = 1 \dots N_T$$

$$F_k^S = \sum_{l \neq k}^{N_T} f_{S_k,S_l} + \sum_{j=1}^{N_T} f_{S_k,T_j} + \sum_{i=1}^{N_V} f_{S_k,V_i} \quad for\ k = 1 \dots N_S$$

To resolve the overlapping conditions, the tumor and stromal cells are relocated following the overdamped spring equation, where *v* is the viscosity of the surrounding medium:

$$\frac{dT_j}{dt} = \frac{1}{v}F_j^T \ and\ \frac{dS_k}{dt} = \frac{1}{v}F_k^S.$$

We apply these equations iteratively to all overlapping tumor and stromal cells until the equilibrium is reached, where $F_j^T = F_k^S = 0$ (**Fig A** in **S1 Text**). The vessels are not subject to relocation, and we allow the vessels to overlap with other vessels to represent different vascular shapes observed in tissue histologic samples that result from the angle at which the tissue slices are cut. This algorithm is only used in the initial phase to create the tissue and thus the same spring stiffness is used for all repulsive forces. Once the overlapping conditions are resolved for all cells, the repulsive forces are deactivated. The cells and vessels are immobile during the oxygen fluctuation period, and cell proliferation and death are neglected during the simulated 30-minute period.

## Oxygen distribution

Oxygen distribution within the tissue $\gamma(\boldsymbol{x},t)$ depends on its influx from vessels, diffusion through the tissue, and the uptake by both stromal and tumor cells. Influx is determined by the location of each individual vessel $\boldsymbol{V}_i$ and the influx rate $\delta_V(t)$, which can vary over time. The influx rate represents a fraction ($0 \leq \delta_V(t) \leq 1$) of the maximum oxygen supply $\gamma_{max}$ characteristic of the oxygen content in the vessels of a given radius $R_V$. The transport through the interstitial space of the tumor tissue is assumed to have a constant diffusion coefficient $\mathcal{D}_\gamma$. The oxygen uptake is defined by the Michaelis-Menten equation with a Michaelis constant $\kappa_m$ common for both tumor and stromal cells, the constant maximum uptake rate for stromal cells $S_{max}$, and the maximum uptake rate for tumor cells $\delta_T(t)T_{max}$ for which the rate ($\delta_T(t) \geq 0$) can change over time. The oxygen kinetics is modeled using the following continuous reaction-diffusion equation:

$$\frac{\partial \gamma(\boldsymbol{x},t)}{\partial t} = \underbrace{\sum_{i=1}^{N_V} \delta_V(t)\gamma_{max}\chi_{R_V}(\boldsymbol{x},\boldsymbol{V}_i)}_{influx} + \underbrace{\mathcal{D}_\gamma\Delta\gamma(\boldsymbol{x},t)}_{diffusion} - \underbrace{\sum_{j=1}^{N_T} \frac{\delta_T(t)T_{max}\gamma(\boldsymbol{x},t)}{\kappa_m + \gamma(\boldsymbol{x},t)}\chi_{R_T}(\boldsymbol{x},\boldsymbol{T}_j)}_{uptake\ by\ tumor\ cells} - \underbrace{\sum_{k=1}^{N_S} \frac{S_{max}\gamma(\boldsymbol{x},t)}{\kappa_m + \gamma(\boldsymbol{x},t)}\chi_{R_S}(\boldsymbol{x},\boldsymbol{S}_k)}_{uptake\ by\ stromal\ cells}$$

Interactions between oxygen defined on the Cartesian grid $\boldsymbol{x} = (x,y)$ and the tumor cells, stromal cells and vessels defined on the Lagrangian grid $\boldsymbol{X} = (X,Y)$ are specified by the indicator function, $\chi_R(\boldsymbol{x},\boldsymbol{X})$, with the interaction radius $R$:

$$\chi_R(\boldsymbol{x},\boldsymbol{X}) = \begin{cases} 1 & \|\boldsymbol{x}-\boldsymbol{X}\| < R \\ 0 & otherwise \end{cases}$$

To generate the initial numerically stable oxygen distribution, the oxygen influx rate for each vessel was set up to the maximum vascular level ($\delta_V(t) = 1$) and the tumor cellular uptake was set up to the base uptake level ($\delta_T(t) = 1$). However, in order to generate oxygen fluctuations in the whole tissue, one or both of these rates were varied. The rates are based on experimentally observable changes in red blood cell flux in tumor vasculature[4] and changes in oxygen uptake by tumor cells grown in different microenvironmental conditions[14,15]. The values of all other model parameters are listed in **Table 1** and more information is provided in **S1 Text**.

## Results

The goal of this study is to identify the possible mechanisms leading to fast (within 3-minute duration) fluctuations in oxygen level observed in *in vivo* tumors. To do that, we first generated a collection of tumor tissues that varied in their vascular and cellular fractions (**section 3.1**) and determined the numerically stable oxygen distribution within these tissues (**section 3.2**). Next, we identified *in silico* tissues for which the average pO$_2$ values best fit the four

**Table 1. Model physical and computational parameters.**

| Physical parameters | Value | References |
|---|---|---|
| Tumor cell diameter | $2R_T = 15\ \mu m$ | [16,17] |
| Stromal cell diameter | $2R_S = 7.5\ \mu m$ | [18] |
| Vessel diameter | $2R_V = 40\ \mu m$ | [19] |
| Force stiffness | $\mathcal{F} = 50\ \mu g/\mu m \cdot s^2$ | [20,21] |
| Medium viscosity | $v = 250\ \mu g/\mu m{\cdot}s$ | [22] |
| Oxygen diffusion | $\mathcal{D}_\gamma = 100\ \mu m^2/s$ | [23,24] |
| Vascular level of oxygen | $\gamma_{max} = 60\ \sigma g/\mu m^3\ [60\ mmHg]$ | [6,19,25] |
| Michaelis constant | $\kappa_m = 134\ \sigma g/\mu m^3$ | [26] |
| Tumor base oxygen uptake rate | $T_{max} = 0.382\ \sigma g/s\ ^*$ cell volume | [26] |
| Stromal base oxygen uptake rate | $S_{max} = 0.382\ \sigma g/s\ ^*$ cell volume | [26] |
| Influx rate | $0{\leq}\delta_V(t){\leq}1$ | |
| Uptake rate | $0{\leq}\delta_T(t){\leq}50$ | |
| **Computational parameters** | | |
| Domain size | $\Omega = [-500,500]\times[-500,500]\ \mu m^2$ | |
| Grid width | $\Delta x = 5\ \mu m$ | |
| Time step | $\Delta t = 0.05\ s$ | |
| Scaling parameter | $\sigma g = 0.5\times10^{-19}g = 0.05\ ag$ | |

specific experimental data (**section 3.3**). These tissues were used to investigate whether the observed short-term oxygen fluctuations can arise as an effect of altered oxygen supply from the vasculature or altered oxygen uptake by tumor cells (**section 3.4**). The identified optimal influx/uptake schedules were then applied to a larger sample of *in silico* tissues with initial pO$_2$ levels close to the experimental data to assess the schedules' robustness and reproducibility (**section 3.5**).

## Generation of *in silico* tissues with a stable oxygen distribution

In general, the oxygen distribution within the tissue—that is, the extent and localization of well-oxygenated vs. hypoxic regions—depend on the number and placement of vessels and both tumor and stromal cells. Therefore, we generated a collection of *in silico* tissues with different vascularity (a fraction of the tissue that is occupied by the vessels), as well as tumor and stromal cellularity (a fraction of the tissue populated by tumor or stromal cells, respectively). In particular, we considered tissue vascularity to be between 0.5% and 5% of the whole tissue area, in increments of 0.5%. The fractions of tissue inhabited by tumor cells was varied between 10% and 95%, and by stromal cells between 5% and 95%, both in increments of 5%. We have ensured that the total of cellular and vascular fractions does not exceed 100% of the tissue area. The locations of all vessels and cells were chosen randomly, and we applied the repulsive force algorithm described above to resolve overlaps between the cells and vessels. In all of these simulations, the oxygen influx from vessels was assumed identical (with $\delta_V(t) = 1$), as was the oxygen uptake by each tumor cell (with $\delta_T(t) = 1$).

For each *in silico* tissue, we applied the diffusion-reaction equation of oxygen kinetics to generate a stable oxygen distribution. As a stability criterion, we calculated the L$_2$-norm between two consecutive oxygen distributions:

$$\varepsilon_\gamma^n = \|\gamma^n - \gamma^{n-1}\|_2 = \sqrt{\sum_{i,j=1}^{N_i,N_j}(\gamma_{ij}^n - \gamma_{ij}^{n-1})^2}$$

where, $\gamma_{ij}^n = \gamma(\mathbf{x}_{ij}, t_n)$ is the oxygen value at the grid point $\mathbf{x}_{ij}$ at time $t_n = t_0 + n\Delta t$, and $N_i \cdot N_j$ are the total number of grid points. A numerically stable oxygen distribution was achieved when the normalized error reached a small enough value ($\overline{\overline{\varepsilon_\gamma^n}} \le 10^{-10}$), where

$$\overline{\overline{\varepsilon_\gamma^n}} = \frac{\varepsilon_\gamma^n}{N_i \cdot N_j}$$

An example of the numerically stable oxygen distribution is shown in **Fig 2**. The tissue morphology with 3.5% vascularity, tumor cell fraction of 55%, and stromal cellularity of 30% is presented in **Fig 2A**, and the final oxygen distribution in **Fig 2B**. The initial oxygen level was set up to $\gamma_{ij}^0 = 0$ mmHg uniformly in the whole tissue domain. The average oxygen level in the whole tissue stabilized at the level of 29.89 mmHg in about $2\text{x}10^4$ iteration steps, reaching a normalized error of $9.99\text{x}10^{-11}$ (**Fig 2C**). The final stabilized oxygen level is independent of the oxygen concentration chosen to initiate this process (**Fig B** and **Table A in S1 Text**).

## Classification of tissues with specific saturation levels

The generated library contains 1,530 tumor tissues of different morphologies with a numerically stabile oxygen distribution. The minimum and maximum average $pO_2$ values were achieved at the levels of 2.06 mmHg and 55.55 mmHg, respectively. All tumor tissues were divided into five classes according to their average $pO_2$ (from 0 to 60 mmHg with increments of 12 mmHg).

The parameter space corresponding to each class is shown in **Fig 3A** in a form of a 3D convex hull, that is, the smallest convex set containing all data points from a given class (color-coded in blue, red, orange, yellow and white). Almost 1/3 of all tissues (506 cases) stabilized at a high level of 36–48 mmHg (yellow region). Moderate $pO_2$ levels of 24–36 mmHg were achieved in 340 tissues (orange region). Low $pO_2$ levels of 12–24 mmHg were reached in 284 tissues (red region), and a similar number of tissues (270) had hypoxic levels of 0–12 mmHg (blue region). The smallest number of tissues (130) reached a very high $pO_2$ level, above 48 mmHg (white region). In general, the higher tissue vascularity and lower cellularity, the higher

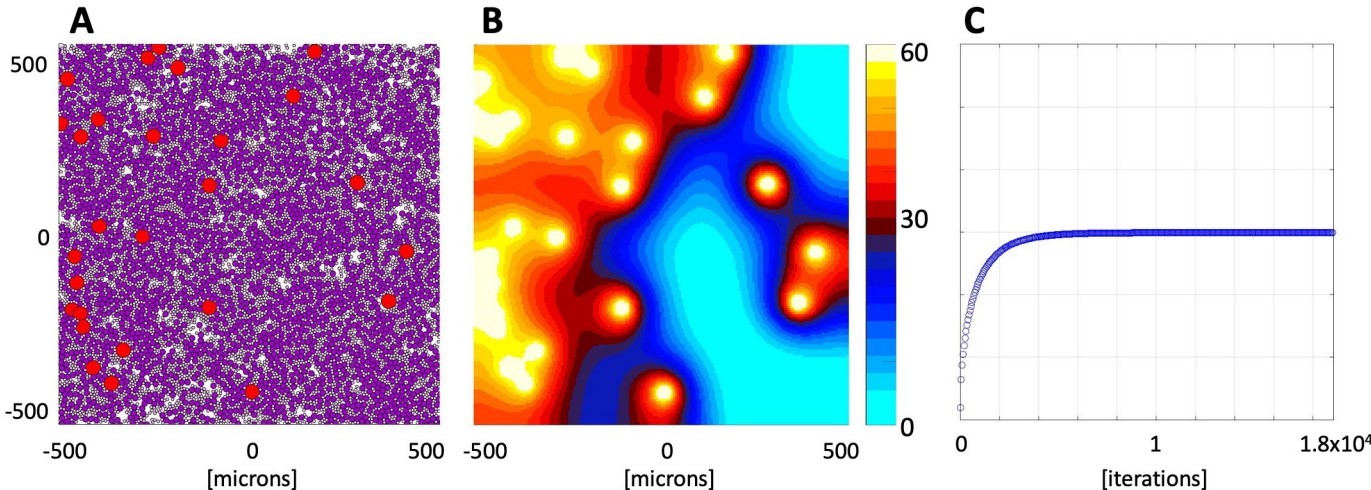

**Fig 2. Stabilized oxygen distribution for an exemplary tissue morphology. A**. *In silico* tissue morphology comprised of 3.5% of vasculature (red circles), 55% of tumor cells (dark purple circles), and 30% of stromal cells (light pink circles). **B**. The stabilized oxygen distribution color-coded using the EPR imaging color scheme, with high oxygen levels (yellow) near the vessels and low oxygen levels (cyan) in poorly vascularized regions. **C**. Changes in the average oxygen concentration over time from initial 0 mmHg to the stable level of 29.89 mmHg.

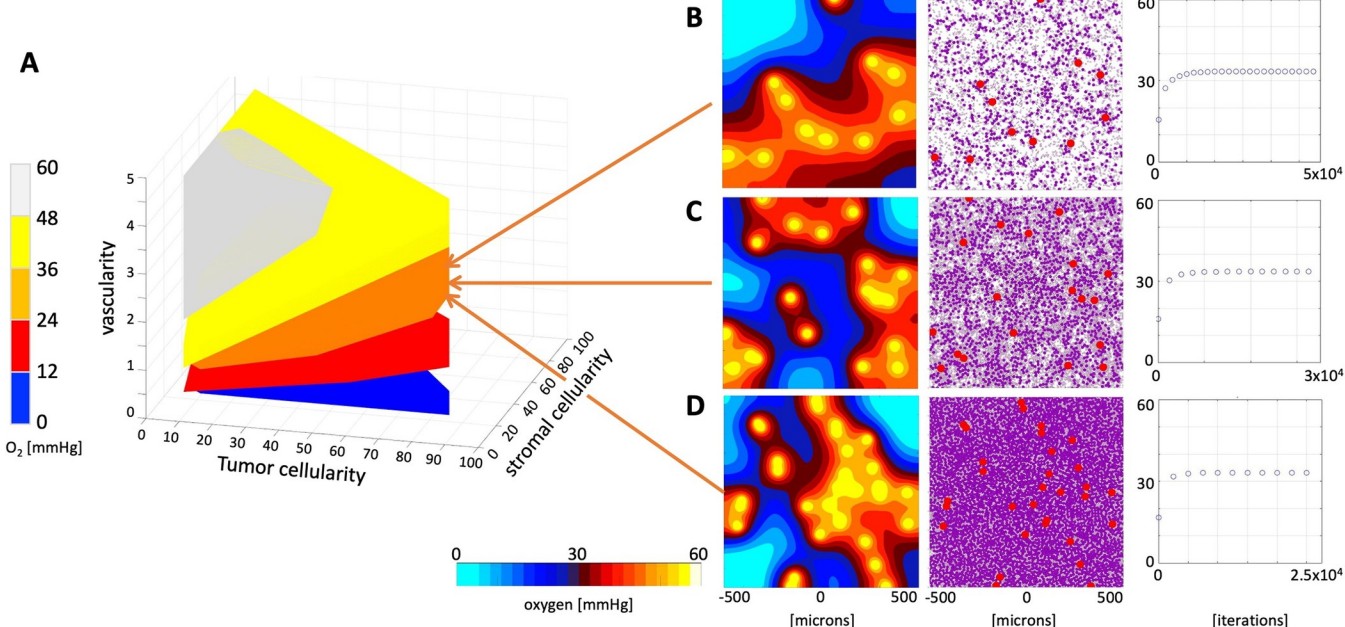

**Fig 3. Classification of tissue oxygenation. A**. A parameter space (convex hulls) of tissues characterized by vascularity, tumor cellularity and stromal cellularity classified into five classes with respect to the stabilized average oxygen level. For every tissue characteristic only one tissue morphology was included. **B-D.** Three examples of tissues with similar oxygen saturation levels: **B.** A tissue with vascularity 1.5%, tumor cellularity 15%, stromal cellularity 15%, and stable oxygen of 33.43 mmHg. **C.** A tissue with vascularity 2.5%, tumor cellularity 30%, stromal cellularity 35%, and stable oxygen of 33.53 mmHg. **D.** A tissue with vascularity 4%, tumor cellularity 75%, stromal cellularity 20%, and stable oxygen of 33.16 mmHg.

level of average tissue $pO_2$. However, the vascularity and cellularity values need to be tightly balanced to achieve a desired level of $pO_2$, which is illustrated by three examples in **Fig 3B–3D**. Each tissue reached an average $pO_2$ level near 33 mmHg, although the increased tissue vascularity (from 1.5% to 2.5%, to 4%) is accompanied by increased total tissue cellularity (from 30% to 65%, to 95%). The time required for the $pO_2$ levels to numerically stabilize is different in each case. Less dense tissues require more time (**Fig 3B–3D** last column), since it takes longer for the oxygen to reach sparsely located cells. Thus, initial changes in oxygen spatial distribution result mostly from diffusion, before cells start consuming oxygen contributing to the influx-outflux balance. We also analyzed how the numerically stabilized levels of oxygen depend on the random locations of vessels and cells by generating 25 different tissues with vascularity and cellularity corresponding to those in **Fig 3B–3D**. They stabilized at the levels of 32.63, 29.98, and 36.15 mmHg, respectively, with a standard deviation of 2–3 mmHg (**Fig C** in **S1 Text**).

## Selection of tissues best fitted to experimental data

For further analysis, we selected computational tissues with stabilized oxygen levels that closely matched the maximum experimental measurement recorded in each of the four ROIs from **Fig 2** in[6]. In particular, region #1 (black) has an initial $pO_2$ of 36.312 mmHg, and the generated tissue with the $pO_2$ closest to it, 36.315 mmHg, contains a vascular fraction of 4%, tumor cell fraction of 15%, and stromal cell fraction of 75% of the tissue area (**Fig 4A**). The experimental region #2 (red) has a maximum $pO_2$ of 22.89 mmHg, and our *in silico* tissue configuration with a vascular fraction of 2.5%, tumor fraction of 45%, and stromal fraction of 40% reached oxygenation of 22.22 mmHg (**Fig 4B**). The third experimental region (blue) has a $pO_2$

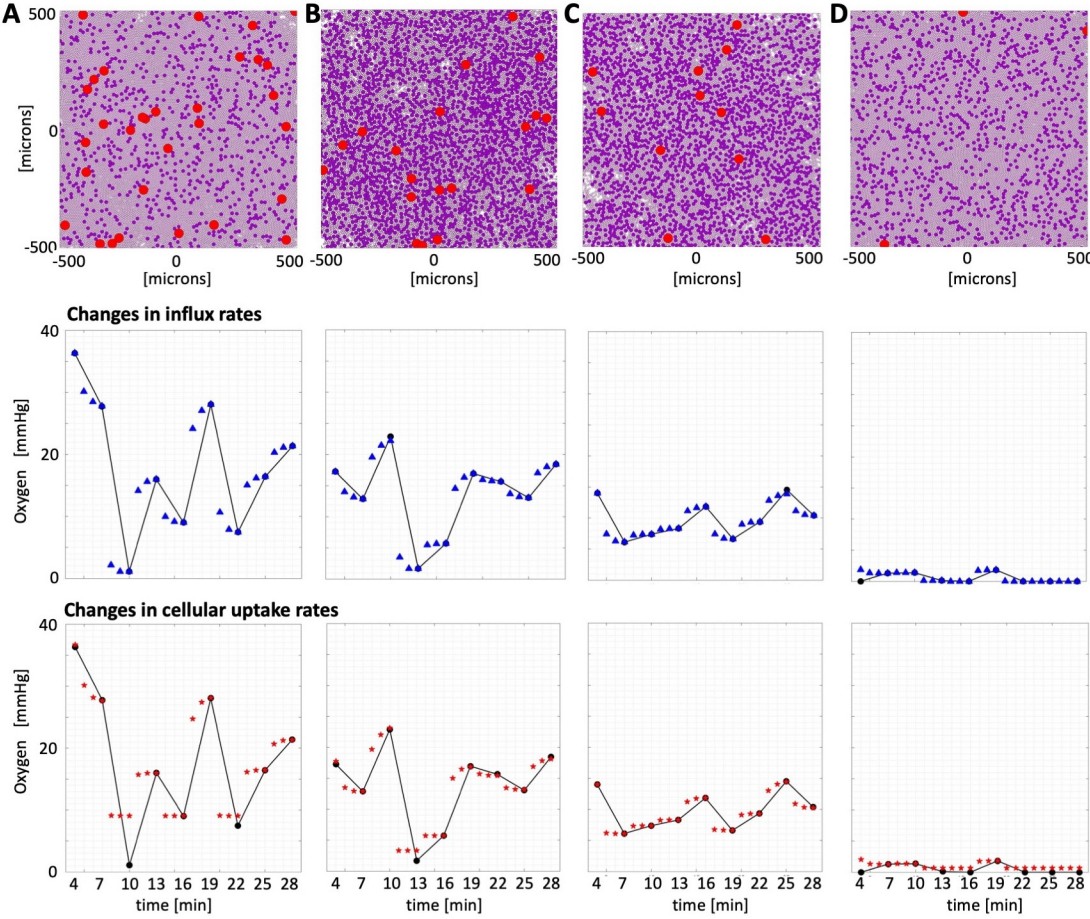

**Fig 4. Reconstruction of oxygen fluctuations in ROIs #1–4.** Tissue configurations (top row) for which numerically stabilized oxygen distributions matched the maximum average pO$_2$ level in each of the region of interests and the reconstructed pO$_2$ fluctuations when either vascular influx rates (middle row) or cellular uptake rates (bottom row) were varied. Straight lines connect experimental data recorded every 3 minutes. Blue triangles (top, influx) and red stars (bottom, uptake) denote computational data recorded each minute. Tissue characteristics: **A.** ROI#1 (black): vascularity 4%, tumor cellularity 15%, and stromal cellularity 75%. **B.** ROI#2 (red): vascularity 2.5%, tumor cellularity 45%, and stromal cellularity 40%. **C.** ROI#3 (blue): vascularity 1.5%, tumor cellularity 40%, and stromal cellularity 45%. **D.** ROI#4 (magenta): vascularity 0.5%, tumor cellularity 20%, and stromal cellularity 75%.

of 13.99 mmHg, while the computational tissue with a numerically stabilized oxygen level of 14.01 mmHg contains a vascular fraction of 1.5%, tumor fraction of 40%, and stromal fraction of 45% of the tissue area (**Fig 4C**). Finally, the fourth experimental region (magenta) has a maximum pO$_2$ near 1.83 mmHg, and the closest generated configuration has an oxygenation level of 2.06 mmHg and a vascular fraction of 0.5%, tumor fraction of 20%, and stromal fraction of 75% of the tissue area (**Fig 4D**).

For each case, our goal was to reproduce the oxygen fluctuations shown in **Fig 2** from [6]. Our approach was to alter either the oxygen influx rates or the oxygen cellular uptake rates every three minutes to match each data point that was recorded in *in vivo* experiments. Since the fluctuations in the four selected regions have different magnitudes, this would allow us to assess whether the given mechanism may be responsible for the observed changes in tissue oxygen levels. Once these rates were determined for the selected *in silico* tissues, we applied the same schedules to a set of tissues that stabilized at the similar oxygen levels to show how robust these optimal schedules are in reproducing oxygen fluctuations.

### Reconstruction of experimentally measured oxygen fluctuations

To test the impact of vascular supply on tissue oxygenation, we simultaneously adjusted the vascular influx of oxygen in each vessel by assigning a fraction $\delta_V(t)$ (between 0 and 1) of the default maximum influx value $\gamma_{max}$. To test the role of tumor cell metabolism on tissue oxygenation, we adjusted cellular uptake in all tumor cells by multiplying the default absorption value $T_{max}$ by a constant ratio $\delta_T(t)$ (between 0 and 50) to account for either decreased or increased cellular uptake. The rates $\delta_V(t)$ and $\delta_T(t)$ were determined using the Mesh Adaptive Direct Search (MADS) method as implemented in MATLAB® by the *patternsearch* routine [27]. This optimization algorithm utilizes a value calculated at a given time point by a simulation of the underlying deterministic system and does not require derivatives of the objective function.

Our optimization goal was to minimize the difference between the average tissue $pO_2$ level recorded experimentally, $\overline{\gamma^E(t_k)}$, and the one computed by our model, $\overline{\gamma^C(t_k)}$, at the end of each 3-minute interval (i.e., for $t_k \in \{4, 7, 10, 13, 16, 19, 22, 25, 28\}$ minutes, $N = 9$ intervals in total), where $x(t_{k-1})$ is a value of either the vascular influx rate $\delta_V(t_{k-1})$ or the tumor cell uptake rate $\delta_T(t_{k-1})$ at the beginning of each 3-minute time interval (i.e., for $t_{k-1} \in \{0, 4, 7, 10, 13, 16, 19, 22, 25\}$ minutes):

$$min \sum_{k=1}^{N} |\overline{\gamma^E(t_k)} - \overline{\gamma^C(t_k)}|$$

$$x = \{x(t_0), \cdots, x(t_{N-1})\}$$

Once the final optimal schedule (influx or uptake) is determined, the normalized $L_2$-norm $\overline{\overline{\mathcal{L}_2}}$ between the simulated and experimental data points is reported as an indication of the goodness of fit (GoF) of the optimal schedule:

$$\overline{\overline{\mathcal{L}_2(\gamma^{opt})}} = \frac{1}{N} \sqrt{\sum_{k=1}^{N} (\overline{\gamma^E(t_k)} - \overline{\gamma^{opt}(t_k)})^2}$$

The optimal influx and uptake rates together with the normalized $L_2$-norms for all ROIs are listed in the **Table 2**. Method convergence for the cases with best and worst GoF (both from ROI#1) is shown in **Fig D** in **S1 Text**.

**Table 2. Influx and Uptake Schedules for four considered ROIs.**

| | Time intervals | | | | | | | | | GoF |
| | 0–4 | 4–7 | 7–10 | 10–13 | 13–16 | 16–19 | 19–22 | 22–25 | 25–28 | $\overline{\overline{L_2(\gamma^{opt})}}$ |
|---|---|---|---|---|---|---|---|---|---|---|
| **Region #1 Black** | | | | | | | | | | |
| Influx $\delta_V(t)$ | 1 | 0.832 | 0.096 | 0.592 | 0.410 | 0.857 | 0.363 | 0.602 | 0.711 | 0.0013 |
| Uptake $\delta_T(t)$ | 1 | 4.625 | 50 | 15.75 | 50 | 4.125 | 50 | 14.875 | 9 | 0.9078 |
| **Region #2 Red** | | | | | | | | | | |
| Influx $\delta_V(t)$ | 0.84 | 0.709 | 1 | 0.1875 | 0.432 | 0.850 | 0.799 | 0.7168 | 0.887 | 0.0754 |
| Uptake $\delta_T(t)$ | 1.875 | 3.125 | 0.875 | 50 | 14.25 | 1.875 | 2.25 | 3 | 1.165 | 0.1986 |
| **Region #3 Blue** | | | | | | | | | | |
| Influx $\delta_V(t)$ | 1 | 0.598 | 0.676 | 0.729 | 0.908 | 0.629 | 0.787 | 1 | 0.832 | 0.0757 |
| Uptake $\delta_T(t)$ | 1 | 6.375 | 4.375 | 3.375 | 1.5 | 5.375 | 2.625 | 0.875 | 2.125 | 0.0779 |
| **Region #4 Magenta** | | | | | | | | | | |
| Influx $\delta_V(t)$ | 0 | 0.738 | 0.762 | 0.121 | 0 | 0.926 | 0 | 0 | 0 | 0.0003 |
| Uptake $\delta_T(t)$ | 50 | 6.875 | 6 | 50 | 50 | 2 | 50 | 50 | 50 | 0.1769 |

The resulting simulated fluctuations are shown in **Fig 4**, and the exemplar oxygen distributions at each stage are shown in **Fig D** in **S1 Text**. The presented results indicate that changes in vascular influx can reproduce experimentally observed fluctuations in $pO_2$ levels in all four ROIs (the normalized $L_2$-norms are below 0.1 in all cases). In the case of ROI#3, these influx alterations are moderate, not exceeding 40%. However, the sudden drops in $pO_2$ level observed in ROI#1 and ROI#2 requires a substantial decrease in oxygen influx, as much as 80% and 99%, respectively. In the case of ROI#4, when the $pO_2$ level reaches a value near 0, the oxygen influx must be completely shut down. These changes in intravascular oxygen content are physiologically plausible, as cases with low or even zero arterial oxygen supply (anoxemia) have been observed[28]. Here, we showed that short-term fluctuations in tissue oxygenation can be achieved by temporal alterations in intravascular oxygen supply.

The changes in tumor cell metabolisms (modeled as an increase in oxygen uptake) can explain smaller fluctuations in tissue oxygenation (ROI#3, the normalized $L_2$-norm below 0.1). It required up to 6-fold changes in the cellular uptake rate to match these fluctuations. This mechanism can also fit cases with near-zero oxygen depletion in the whole tissue patch (ROI#4 and ROI#2, with $L_2$-norms near 0.2). However, it failed to reproduce large (more than 5-fold) and rapid fluctuations (ROI#1, with normalized $L_2$-norm near 1), even considering changes in cellular uptake of up to 50-fold. Thus, in general, oxygen fluctuations were not captured by changes in cell metabolism.

## Robustness of optimal schedules

The optimal influx/uptake schedules described above were determined using four particular tissues for which the average oxygen level has stabilized at values closest to the maximum value recorded for each of the four ROIs from **Fig 2** in[6]. Here, we investigated whether these optimal schedules applied to other tissues will reproduce oxygen fluctuations recorded experimentally. One motivation was to test whether tissues of various morphologies but similar average $pO_2$ levels will respond in a similar way to the schedules that were optimized using one of these tissues. If fluctuations in the average $pO_2$ level are not sensitive to tissue morphology, any of these tissues or a small subset of these tissues can be used for further simulation studies of diffusive therapeutic agents and their impact on tumor progression. Another motivation was to provide a link between the average data value recorded for radiologic image voxels and the structure of the corresponding tissues. Potentially, a very large number of tissue structures may result in the same average $pO_2$ level. Our goal here was to investigate whether additional information, such as temporal data recorded for the same voxel, will result in a reduced number of tissue morphologies that reproduce that data. If this tissue number is smaller, we can determine better conditions for selection of tissue morphologies for further studies of intratumoral drug or biomarker distribution. Taken together, we can identify which mechanisms can reproduce the experimentally observed fluctuations, and to provide criteria for selection of different tissues for which these fluctuations can be reproduced.

To achieve our goals, we first identified four sets of tissues with oxygen levels that stabilized within +/- 3.5 mmHg of the maximum value recorded for each ROI (#1-black, #2-red, #3-blue, and #4-magenta). The number of such representative tissues is listed in **Table 3** for each ROI. Next, we applied the optimal influx schedules that were determined for each ROI to the corresponding set of representative tissues. Separately, we also applied the optimal uptake schedules to these tissues. To measure how well the applied schedules reproduced the experimentally observed oxygen fluctuations, we recorded the normalized $L_2$-norms ($\overline{\overline{\mathcal{L}_2(\gamma^{opt})}}$), as described above. The total number of tissues considered and the numbers of tissues with $L_2$-norms below a threshold value of 0.2 are listed in **Table 3** for each ROI.

**Table 3. Number of tissues representing each ROI and tissues fitting each fluctuation.**

| Number of representative tissues: | ROI#1 | ROI#2 | ROI#3 | ROI#4 |
|---|---|---|---|---|
| within +/- 3.5 mmHg from experimental value | 277 | 189 | 147 | 123 |
| with $L_2$-norm<0.2 for the influx schedule | 42 | 38 | 40 | 61 |
| with $L_2$-norm<0.2 for the uptake schedule | 0 | 6 | 15 | 73 |

The obtained results are also summarized graphically in **Fig 5**. The convex hulls represent a range of tissue characteristics (i.e., vascularity, tumor cellularity, and stromal cellularity) that satisfy a particular condition under consideration. The cyan convex hulls in **Fig 5A** represent all tissues with the oxygen level that stabilized within +/- 3.5 mmHg of the maximum value recorded for each ROI. A subset of tissues for which the optimal influx schedule resulted in oxygen fluctuations that fitted the experimental data with a normalized $L_2$-norm below 0.2 are shown as green convex hulls. Similarly, a subset of tissues for which the optimal uptake schedule fitted the experimental data with the $L_2$-norm below 0.2 are shown in black. The scatter plots in **Fig 5B** show the relationship between the initial numerically stable $pO_2$ (the x-axis shows the deviation of the simulated $pO_2$ level from the experimental measurement) and the normalized $L_2$-norm value for each tissue (y-axis). The green dots represent data for the optimal influx schedule and grey dots show data for the optimal uptake schedule. The red dashed lines represent the normalized $L_2$-norm value of 0.2.

In general, the closest the simulated stabilized oxygen level is to that measured experimentally, the more successful the optimal influx schedule is, since all green dots located below the red threshold line are concentrated near the 0 value in **Fig 5B**. These tissues are also co-localized in **Fig 5A**, although the tissue characteristics shown as the green convex hulls span a broader range of values. The exception is ROI#4, for which almost all considered tissues responded to influx and uptake schedules by following the experimental fluctuations (the

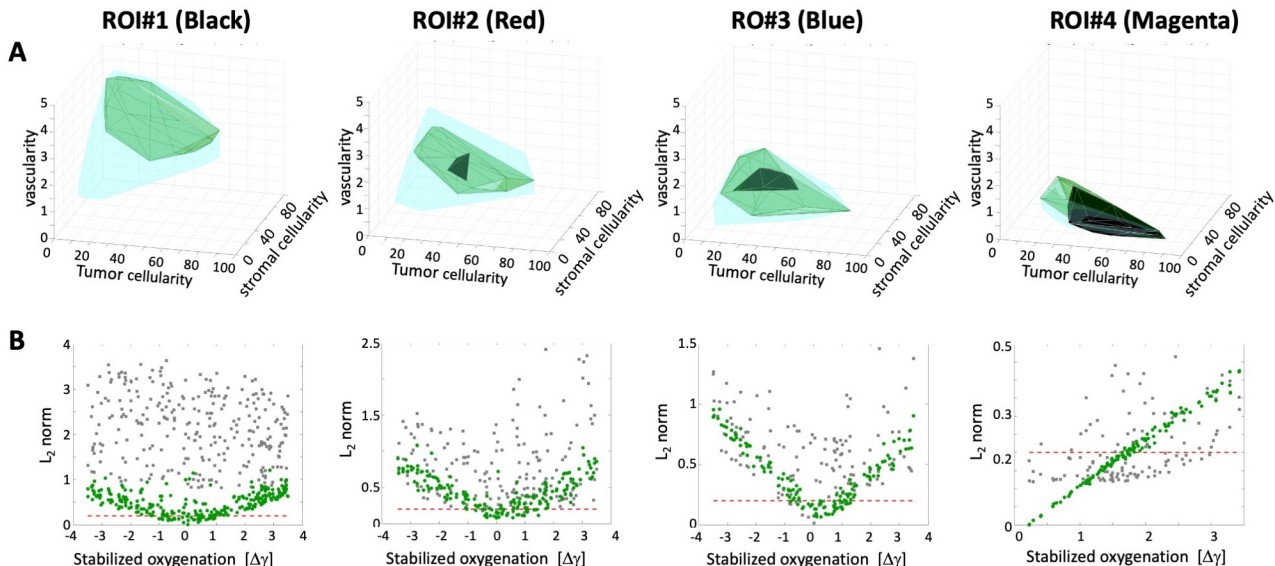

**Fig 5. Robustness of optimal influx and uptake schedules. A.** Parameter spaces of all tissues with oxygen levels within +/- 3.5 mmHg of the maximum experimental value for each ROI; 3D convex hulls shown in cyan, together with convex hulls for optimal influx schedule (green) and optimal uptake schedule (black) that fit experimental data with normalized $L_2$-norm smaller than 0.2. **B.** Normalized $L_2$-norms for influx schedule (green dots) and uptake schedule (grey dots) for each tissue from the cyan convex hull. The red dashed line represents the $L_2$-norm value of 0.2. The results are shown from left to right for: ROI#1 (black), ROI#2 (red), ROI#3 (blue), and ROI#4 (magenta).

green and black convex hulls almost overlap with the cyan one). The fluctuations in this region were very small, so they were easier to reproduce by both schedules. As oxygen fluctuations increased (from ROI#3 to ROI#2, to ROI#1), the numbers of representative tissues that followed the experimental data was decreased, and there was no tissue in ROI#1 for which the optimal cellular uptake schedule resulted in fitting with the normalized $L_2$-norm below 0.2 (no black convex hull).

Several interesting observations can be made based on these results. We showed that only a fraction of the tissue morphologies with oxygen levels that stabilized near the given experimental data are able to reproduce temporal changes in tissue $pO_2$ when the vascular influx of oxygen is varied. These well-fitted tissues have numerically stabile $pO_2$ levels within +/- 1mmHg of the experimental data. Thus, for future applications we can reduce the searching radius from 3.5 to 1 mmHg in order to find plausible tissue morphologies.

By comparing simulation results with the vascular influx schedule vs. the cellular uptake schedule, we conclude that alterations in vascular oxygen levels were able to reproduce the observed fluctuations. On the other hand, in order to achieve the same effect when the metabolic changes in tumor cells are considered, the cells would need to increase their oxygen absorption by 50-fold over a span of 3 minutes, which may not be biologically feasible. While it has been reported in the literature that cellular oxygen uptake can vary greatly between cell lines (i.e., 1–350 amol/s per cell[29], and 1–120 amol/s per cell[30]), changes reported in the same cells varied no more than 10-fold when the culture conditions were modified[14,31].

## Discussion

Motivated by published experimental data that showed frequent fast fluctuations in tissue oxygenation (as high as 30 mmHg over 3-minute intervals), we investigated plausible biological mechanisms that could explain these results. The first hypothesis we tested was that $pO_2$ fluctuations are related to changes in vascular oxygen supply. We generated a large number of in silico tumor morphologies and selected those for which numerically stable $pO_2$ levels were the closest to the experimental data in each region of interest (ROI). Next, we used computational optimization techniques to determine influx schedules that best fitted the experimental fluctuation data. Finally, we applied these optimal schedules to other tissues with similar $pO_2$ levels to test whether they will respond in a similar way to the same influx schedules. This procedure showed that rapid changes in vascular oxygen supply can explain the fluctuations observed in [6] in all considered ROIs.

However, the same mouse experiments showed no differences in the intensity of an EPR-specific imaging tracer in the same ROIs. Since this tracer is supplied intravenously but is not absorbed by the cells, this suggested that blood flux in the tumor vasculature is steady and that the $pO_2$ fluctuations may be a result of increased oxygen uptake by the tumor cells. Therefore, we also tested the hypothesis that modulations in cellular oxygen absorption are responsible for the observed $pO_2$ fluctuations within the tissue. Using the same set of in silico tissues, and applied computational optimization techniques to determine most optimal uptake schedules that fitted the experimental fluctuation data. Again, we applied these optimal schedules to other tissues with similar $pO_2$ levels to test schedule robustness. This procedure showed that in order to fit fluctuations with more than 5-fold magnitude, the 50-fold changes in cellular oxygen absorption would have to occur over 3-minute intervals, which may not be biologically feasible. Thus, we showed computationally that changes in oxygen influx are the most probable explanation of cyclic hypoxia observed in EPR imaging experiments reported in[6].

Moreover, we provided a link between the average data value recorded by radiologic images and the cellular/vascular architecture of the tumor tissue. Since a cross section of a single EPR

voxel has an area about a millimeter square, the underlying tissue patch is large enough to contain subregions of different characteristics. For example, it may include zones with no vasculature, with severe hypoxia, or high cellular density that is poorly penetrated by drugs, or may harbor resistant tumor cell subpopulations. Such cellular-scale phenomena will not be reflected in the average values reported by radiologic images. To capture information on the cellular scale based on average values novel computational approaches are needed. Potentially, a very large number of tissue architectures may result in the same average $pO_2$ level. However, by adding a series of temporal data recorded for the same voxel, we showed that the number of tissue morphologies capable of reproducing the underlying temporal dynamics of oxygen is greatly reduced. In the cases discussed here, we were left with about 40 different tissues for each ROI out of over 1,500 distinct tissues generated initially. This remaining number of tissues is computationally manageable for any further analysis or simulations, giving us a tool for microscale modeling (at the individual cell level) based on macroscale data (average value in tissue voxel).

The model presented here was developed with some simplified assumptions. All vessels in the model are of the same size and contain identical levels of oxygen. Similarly, all tumor cells are of the same size, as are the stromal cells. This was done to reduce model complexity, since the spatial variations in tissue morphology (vascular, tumor and stromal fractions) already introduced a fair amount of heterogeneity. However, in future research, we will incorporate variable sizes of structural tissue elements, as well as non-uniform oxygen influx and consumptions rates in tumor and various stromal cells. It was discussed in[6] that oxygen fluctuations are correlated with local vascular functionality within the tumor. While currently this is not included, in future applications we can add a possibility to model whether the vasculature is patent. In addition, the voxels of radiologic images are volumetric, and we only model the 2D voxel cross section here. Our model can be adapted to full 3D space and can incorporate 3D spatial heterogeneities. All mathematical equations, both the agent-based model rules and the reaction-diffusion equation for oxygen kinetics, can be easily extended to a 3D space. Tumor and stromal cells can be represented as spheres (as we have done in[11,32] and others were reviewed in[33,34]), and tumor vasculature as collections of branched tube segments (as in[35,36,37] and in models reviewed in[33,34]). The only difficulty in 3D models is in visualization of irregular patterns of diffused oxygen and in the time required to complete these simulations, since smaller time steps may be needed to assure computational code stability and convergence.

To our knowledge, this is the first model that generates tissue morphologies corresponding to voxels of radiologic images. The topic of connecting radiology and histology images is of increasing interest since cellular-resolution images harbor detailed information about tumor composition, spatial heterogeneities, and molecular or metabolic landscapes. However, histology samples are collected from tumor biopsies or resections, and thus can only be acquired a limited number of times. In contrast, radiologic images provide non-invasive longitudinal information of tumor states and responses to treatments. Several co-registration methods between these two imaging modalities have recently been developed[38–40]. While mathematical modeling has been previously used in connection with radiologic images[41–46], only continuous models were utilized. Several different agent-based models were, in turn, used to model analogues of in vivo tumor histology[47–50], but were not extended to radiologic imaging data. Fluctuations in the tumor microenvironment have been addressed in the context of long-term evolutionary dynamics, but have not compared to experimental data[51,52].

While we used pre-clinical data collected using EPR imaging here, our goal for future studies is to develop a similar macro-scale to micro-scale model based on clinically-relevant radiologic imaging. Several different minimally invasive imaging tests can be used for longitudinal

monitoring of tumor response to therapies, such as computed tomography (CT), magnetic resonance imaging (MRI), or positron emission tomography (PET). Of particular interest to our studies are methods that can provide additional information related to tissue structure or oxygen/drug pharmacokinetics. The dynamic contrast-enhanced MRI (DCE-MRI) can capture temporal information on tissue perfusion, microvascular permeability, vascular volume fraction, extracellular-extravascular volume fraction, and diffusivity of tissue water, which, combined, can predict outcomes and guide therapy[53,54]. To augment temporal resolution, new under-sampling techniques coupled with parallel imaging methods (GRASP, Golden-angle Radial Sparse Parallel imaging) can be used to continuously acquire images over a long-term continuum (i.e. 5 minutes) at repeatable, small time intervals (i.e. 2.5 seconds), generating a detailed library of exquisitely time-resolved data[55]. Two other non-invasive imaging approaches, TOLD MRI (Tissue Oxygenation Level Dependent MRI) and BOLD MRI (Blood Oxygenation Level Dependent MRI), can be used to visualize information on tumor oxygenation and vascular hemodynamics[56,57]. This additional information at a voxel level can be used to reduce the number of representative tissue morphologies in our micro-macro scale mechanistic link for cell-scale simulations of various anti-cancer treatments (e.g., chemotherapy, hypoxia-activated targeted therapy, radiotherapy, or immunotherapy). These microscopic-level predictions can be compared to longitudinal radiologic data on a tissue voxel scale. This can give us an insight into heterogeneities of intratumoral drug distributions or immune cell penetration patterns, and on the emergence of tissue subregions that can harbor chemo-, or radio-resistant cells. Thus, the mechanistic models of the tumor, such as the one we have described here, will enable patient-specific simulations to predict the trajectory of tumor response to specific interventions.

## Supporting information

**S1 Text. This supporting information includes the computational procedure for resolving cell overlaps, discussion on the choice of model parameters, description of an algorithm for stabilizing oxygen gradient, and details of the optimization protocol used for fitting experimentally observed fluctuations. Fig A. Resolving cell overlapping conditions using repulsive forces.** A left-top quarter of the tumor tissue domain characterized by 2% vascularity, 30% tumor cellularity, and 35% stromal cellularity with tumor cells represented by purple circles, stromal cells as pink circles, and vessels as red circles. **A.** An initial iteration 0 before repulsive forces are applied, showing overlapping cells and vessels. **B.** By iteration 25, repulsive forces have been applied and cell relocation has begun; only some cells remained overlapped. **C.** In iteration 100, all overlapping conditions have been resolved and cell positions have stabilized to reflect no overlapping. We allow vessels to overlap to represent irregular vessels shapes often seen in histology images.**Fig B. Oxygen stabilization within the tumor tissue and the role of initial oxygen concentration. A-D.** Snapshots showing oxygen distribution during the stabilization process, at iterations 0, 2000, 10000, and 18970. **E.** Temporal evolution of the average oxygen level from 0 mmHg until is stabilizes at the 29.89 mmHg with the stabilization error below $10^{-10}$. **F.** Tissue morphology with 3.5% vascular fraction, 55% tumor cellularity, and 30% stromal cellularity; vessels are represented by red circles, tumor cells by purple circles, and stromal cells by pink circles. **G.** Temporal evolution of average oxygen levels for 21 simulations of the same tissue shown in **F.** Each simulation is indicated by a different color that corresponds to initial uniform tissue oxygenation. All simulations stabilized around the 29.89 mmHg, however, simulations with lower initial oxygen concentrations have stabilized faster (cyan and blue lines) than those that started with higher initial oxygen concentrations (red and yellow lines). **Table A. Initial tissue oxygenation vs. final stabilized oxygen level.** Stabilized

average oxygen levels for tissue of characteristics: 3.5% vascular fraction, 55% tumor cellularity, and 30% stromal cellularity, and initial uniform oxygen concentration reported as % of the maximal value of 60 mmHg. The stabilized oxygen levels units: mmHg. **Fig C. Analysis of oxygen stabilization for tissues of identical characteristics but different morphologies. A**. Levels of oxygen stabilization for 25 tissues of vascularity: 1.5%, tumor cellularity: 15%, stromal cellularity: 15%, with an average oxygenation level of 32.63 mmHg +/- 3.3 mmHg. **B**. Levels of oxygen stabilization for 25 tissues of vascularity: 2.5%, tumor cellularity: 30%, stromal cellularity: 35%, with an average oxygenation level of 29.98 mmHg +/- 2.4 mmHg. **C**. Levels of oxygen stabilization for 25 tissues of vascularity: 4%, tumor cellularity: 75%, stromal cellularity: 20%, with an average oxygenation level of 36.15 mmHg +/- 2.4 mmHg. **D**. Spatial gradient of oxygen with lowest average level among tissues in **A**. **E-I**. Spatial gradients of oxygen with highest (left) and lowest (right) average levels among tissues considered in **A-C**. **Fig D. Convergence of the MABS method for influx and uptake schedules for ROI #1 (black). A**. Experimental fluctuations (solid line) and simulated fluctuations (blue triangles) for the influx schedule with corresponding oxygen distributions at each time point (above the arrows). An inset shows tissue morphology. **B**. The convergence graphs for each of the 8 time segments with the converging influx rates (red pins) and the converging objective function values (blue pins). **C**. Experimental fluctuations (solid line) and simulated fluctuations (red stars) for the uptake schedule with corresponding oxygen distributions at each time point (above arrows). **D**. The convergence graphs for each of the 8 time segments with the converging uptake rates (red pins) and the converging objective function values (blue pins).
(DOCX)

## Author Contributions

**Conceptualization:** Jessica L. Kingsley, James R. Costello, Natarajan Raghunand, Katarzyna A. Rejniak.

**Formal analysis:** Jessica L. Kingsley.

**Funding acquisition:** James R. Costello, Natarajan Raghunand, Katarzyna A. Rejniak.

**Methodology:** Jessica L. Kingsley, Katarzyna A. Rejniak.

**Supervision:** Katarzyna A. Rejniak.

**Visualization:** Jessica L. Kingsley.

**Writing – original draft:** Jessica L. Kingsley, James R. Costello, Natarajan Raghunand, Katarzyna A. Rejniak.

**Writing – review & editing:** Jessica L. Kingsley, James R. Costello, Natarajan Raghunand, Katarzyna A. Rejniak.

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
