## [Decision Letter · Decision Letter 0]

1 Apr 2021

Dear Dr. Rejniak,

Thank you very much for submitting your manuscript "Bridging cell-scale simulations and radiologic images to explain short-time intratumoral oxygen fluctuations" for consideration at PLOS Computational Biology.

As with all papers reviewed by the journal, your manuscript was reviewed by members of the editorial board and by several independent reviewers. In light of the reviews (below this email), we would like to invite the resubmission of a significantly-revised version that takes into account the reviewers' comments.

We cannot make any decision about publication until we have seen the revised manuscript and your response to the reviewers' comments. Your revised manuscript is also likely to be sent to reviewers for further evaluation.

Sincerely,

Philip K Maini

Associate Editor

PLOS Computational Biology

Feilim Mac Gabhann

Editor-in-Chief

PLOS Computational Biology

Reviewer's Responses to Questions

**Comments to the Authors:**

Reviewer #1: review uploaded as an attachment

Reviewer #2: The manuscript by Kingsley et al. presents a computational investigation into the cause of small-scale oxygen fluctuations in solid tumours. An agent-based model is used to generate a set of micro-scale tissue morphologies which emulate voxel-scale O2 gradients observed in squamous cell carcinoma VII (SCCVII) in vivo, using electron paramagnetic resonance imaging (EPRI; data from Yasui et al. (2010)). Using this framework, the authors showed that changes in intravascular O2 reproduce the magnitude of fluctuations in vivo whereas modifying cellular absorption did not.

The written language of the manuscript was at times fragmented which hindered readability but I enjoyed the novelty of generating an agent-based modelling framework to infer plausible explanations of in vivo observations. That said, I have significant reservations regarding the model’s application to EPRI data which may impact the biological conclusions of the manuscript.

In the Supplementary Material the authors state that the “model represents features characteristic of many solid tumors, thus we did not focus on any specific tumor type”. I agree. The modelling framework can, to an extent, represent certain features of solid tumours. However, not focusing on tumour type in this manuscript when attempting to recreate pO2 fluctuations observed by Yasui et al. (2010) (Fig. 1 in Kingsley et al.) in SCVII tumours, I believe, is an error. Tumour microenvironment heterogeneity can vary wildly between tumour types and so by not targeting SCVII tumours may negate the authors' biological inferences.

To mimic the fluctuations and magnitudes in O2 observed by Yasui et al. (2010) in silico, the authors should (where possible) parameterise their models with respect to SCCVII tumours. For example, Kingsley et al. used a constant vessel diameter of 40 microns, yet SCVII have diameters in the range of 14.5 – 11.0 microns for tumours of similar masses [1] (see Fig. 23 in [1]). The authors here also increment tissue vascularity from 0.5 to 5%. However, Yasui et al. (2010) reports vascular density of 27.7 ± 2.1% and 8.1 ± 1.0% for small and large SCVII tumours, respectively. Similarly, [1] reports vascular density as between 2 – 7.5% for large SCVII tumours (see Fig. 25 in [1]). Generating new tissue morphologies with these reported values may not change the authors’ conclusion regarding cycling hypoxia but will clearly alter the range of morphologies which replicate experimentally observed pO2 fluctuations.

In general, the manuscript would benefit from relating its conclusions back to the study of Yasui et al. (2010) in terms of the additional insights the in silico model provides. For example, Kingsley et al. indicate that upstream effects (via varying vessel influx levels) replicate O2 gradients observed in the EPRI data. Comparatively, using analysis of the SCVII tumours via immunohistochemistry, Yasui et al. (2010), pO2 fluctuations correlated with pericyte density (i.e., local vascular functionality) rather than vascular density. Kingsley et al. could use their model to indict whether this hypothesis may be true, whilst highlighting that it does not currently include local vascular functionality but has the scope to in the future.

Presenting the generalised modelling framework and then applying it to the EPRI data to make conclusions regarding SCVII tumours I believe would really elevate the manuscript.

The following are further points which could improve the manuscript:

1. There are several places in the manuscript where I would expect a reference to support a written statement but one is not given. For example, Line 76 regarding oxygen gradients at 120-180 um from the vasculature.

2. The introduction dives straight into discussing the ROIs observed in Yasui et al. (2010). However, a brief explanation of how EPRI works along with spatial information such as EPRI resolution in comparison to microenvironment features in the authors’ tissue morphologies. This would develop the reader’s understanding from the start of the limitations of EPRI and how the manuscript seeks to link the micro- and macro-scales.

3. Lines 126 and 491, voxel dimensions are defined as mm2 – voxels are volumetric.

4. Following (3), the authors do not discuss the fact that the model is 2-dimensional (although not explicitly stated) yet EPRI is 3-dimensional. From Yasui et al. (2010), I assume that the ROIs from are 3-dimensional ROIs and that plot of changes in pO2 (Fig 2c) is an average for a given ROI? This should be stated and highlighted that the current model does not incorporate 3D spatial heterogeneities.

5. The authors discuss “large” fluctuations in tumour oxygenation. It would be useful to write in-text what the authors consider as large (i.e., magnitude).

6. Line 165 to 166 state that vessels can overlap to represent shapes observed in histological slices and that further information of the algorithm is provided in Supporting Information S1. However, the only indication of overlapping vessels is in Fig S3f. Further, examples of overlapping vessels with similar images of these shapes in histological images would be useful. In addition, where vessels with multiple overlaps as in Fig Sf (see 3 overlapping vessels in the bottom left-hand corner) excluded from the library of tissue samples? I imagine these shapes are not observed in histology and so should be omitted.

7. Lines 507 to 534 in the Discussion reads like a literature review of imaging methods to observe tissue oxygenation. It is odd that this forms a significant portion of the Discussion.

8. A discussion of the model limitations of the model / future development (heterogeneous vasculature and O2 influx, 3D etc) the Discussion would be very useful.

9. It would be advisable to use colour blind safe colour schemes for the figures.

10. In Fig 4., the iteration subfigures on the right-hand side may be presented better on a log x-axis.

11. In Fig 5., the subfigures showing oxygen vs. time should be rescale to 0 – 40 mmHg and given a clear colour scheme. Currently, it is difficult to see the data points.

12. Supplementary Figure S5 is far too small to read. Consider splitting and enlarging into several figure panels.

[1] Trotter, M.J. (1990). ‘Intermittent blood flow in the murine SCCVII squamous cell carcinoma’. PhD Thesis. University of British Columbia.

**Have all data underlying the figures and results presented in the manuscript been provided?**

Reviewer #1: **No: **code for reproducing simulations not made available (to the best of my knowledge)

PLOS authors have the option to publish the peer review history of their article (what does this mean?). If published, this will include your full peer review and any attached files.

Reviewer #1: No

Reviewer #2: No

**Have the authors made all data and (if applicable) computational code underlying the findings in their manuscript fully available?**

Reviewer #2: **No: **No code repository provided.
---

## [Decision Letter · Decision Letter 1]

1 Jun 2021

Dear Dr. Rejniak,

Thank you very much for submitting your manuscript "Bridging cell-scale simulations and radiologic images to explain short-time intratumoral oxygen fluctuations" for consideration at PLOS Computational Biology. As with all papers reviewed by the journal, your manuscript was reviewed by members of the editorial board and by several independent reviewers. The reviewers appreciated the attention to an important topic. Based on the reviews, we are likely to accept this manuscript for publication, providing that you modify the manuscript according to the review recommendations.

Sincerely,

Philip K Maini

Associate Editor

PLOS Computational Biology

Feilim Mac Gabhann

Editor-in-Chief

PLOS Computational Biology

[LINK]

Reviewer's Responses to Questions

**Comments to the Authors:**

Reviewer #1: I have read through the revised manuscript and the authors response to the comments I raised. Overall, the authors have done a good job of addressing most of the comments that were raised. However, there are a few places where further clarification is needed. I will list these below.

Comment 2. The authors justify their decision not to include dimensional estimates of the length scales involved in their model by stating that there is no spatial scale provided in the experimental paper on which there work is based. I do not consider this sufficient justification – the authors could, for example, estimate typical length scales either from the existing experimental literature or length scales consistent with the vascular densities that they use in their simulations. I would encourage them to do this, so that the reader more readily understands the physical situations that they are modelling.

Comment 3. The literature review in the discussion does not reflect the range of modelling approaches in the literature. The authors may find the following references of relevance:

• T Roque, L Risser, V Kersemans, S Smart, D Allen, P Kinchesh, S Gilchrist, AL Gomes, JA Schnabel, MA Chappell (2016). A DCE-MRI Driven 3-D Reaction-Diffusion Model of Solid Tumour Growth. IEEE Trans Medical Imaging (DOI:10.1109/TMI.2017.2779811)

• C Villa, MAJ Chaplain & T Lorenzi (2021). Modeling the emergence of phenotypic heterogeneity in vascularized tumors. SIAM J Applied Mathematics, 81(2), 434-453.

Comment 5: please can the authors clarify the relevance of measured changes in pyruvate levels obtained from in vitro experiments. I do not understand how changes in oxygen consumption in response to changes in pyruvate levels provide information about how cells response to changes in oxygen levels. Equally, I respectfully disagree that the impact of changes in oxygen levels on stromal cells has not been studied in the literature. See for example G Solani et al (2011), B Zhao et al (2016).

Comment 6: please clarify what an L2-norm of 10^{-10} means – the significance of this value will depend on the oxygen distributions being measured.

Comment 8: I do not understand why it takes longer for the system to reach a steady state as the density of cells increases. For a domain of the same size, the diffusive timescale will be the same. Does the dominant balance change in some way? The justification here could be further clarified

If the authors can satisfactorily address the above comments, then I would be very happy to recommend their article for publication. It is a great piece of work, improved with the revisions they have made.

Reviewer #2: In the revised manuscript, the authors have provided increased clarity and detail, particularly in the introduction and discussion, which has elevated the manuscript. I would also like to thank the authors on their detailed response particularly with regard morphological heterogeneities. Their comments have satisfied my initial concerns. I appreciate that more tumour-specific functional and architectural heterogeneities can by incorporated in future work, which has been noted in the discussion.

Subject to the minor comments below, I recommend this manuscript for publication and congratulate the authors.

- Lines 132 / 133 – the authors’ write “…too short for significant modifications in oxygen transport”. Is this comment regarding cellular update and interstitial transport?

- This reviewer apologises for their poor eyesight but Supporting Figure S4 is still too small. An alternative to reordering the figure panels may be to increase the resolution of the image in the Word Doc, as it is currently too pixelated even when zooming in.

**Have the authors made all data and (if applicable) computational code underlying the findings in their manuscript fully available?**

Reviewer #1: Yes

Reviewer #2: Yes

PLOS authors have the option to publish the peer review history of their article (what does this mean?). If published, this will include your full peer review and any attached files.

Reviewer #1: No

Reviewer #2: No

Figure Files:

Data Requirements:

Reproducibility:

References:

---

## [Editor Report · Decision Letter 2]

22 Jun 2021

Dear Dr. Rejniak,

We are pleased to inform you that your manuscript 'Bridging cell-scale simulations and radiologic images to explain short-time intratumoral oxygen fluctuations' has been provisionally accepted for publication in PLOS Computational Biology.

Best regards,

Philip K Maini

Associate Editor

PLOS Computational Biology

Feilim Mac Gabhann

Editor-in-Chief

PLOS Computational Biology

---

## [Editor Report · Acceptance letter]

19 Jul 2021

PCOMPBIOL-D-21-00412R2 

Bridging cell-scale simulations and radiologic images to explain short-time intratumoral oxygen fluctuations

Dear Dr Rejniak,

I am pleased to inform you that your manuscript has been formally accepted for publication in PLOS Computational Biology. Your manuscript is now with our production department and you will be notified of the publication date in due course.

With kind regards,

Zita Barta
